

# Eye movement control during visual pursuit in Parkinson's disease

Chia-Chien Wu[1], Bo Cao[2], Veena Dali[1], Celia Gagliardi[1], Olivier J. Barthelemy[3], Robert D. Salazar[3], Marc Pomplun[4], Alice Cronin-Golomb[3] and Arash Yazdanbakhsh[1,3]

[1] Center for Computational Neuroscience and Neural Technology, Boston University, Boston, MA, USA
[2] Department of Psychiatry, Faculty of Medicine & Dentistry, University of Alberta, Edmonton, Alberta, Canada
[3] Department of Psychological and Brain Sciences, Boston University, Boston, MA, USA
[4] Department of Computer Science, University of Massachusetts at Boston, Boston, MA, USA

## ABSTRACT

**Background:** Prior studies of oculomotor function in Parkinson's disease (PD) have either focused on saccades without considering smooth pursuit, or tested smooth pursuit while excluding saccades. The present study investigated the control of saccadic eye movements during pursuit tasksand assessed the quality of binocular coordinationas potential sensitive markers of PD.

**Methods:** Observers fixated on a central cross while a target moved toward it. Once the target reached the fixation cross, observers began to pursue the moving target. To further investigate binocular coordination, the moving target was presented on both eyes (binocular condition), or on one eye only (dichoptic condition).

**Results:** The PD group made more saccades than age-matched normal control adults (NC) both during fixation and pursuit. The difference between left and right gaze positions increased over time during the pursuit period for PD but not for NC. The findings were not related to age, as NC and young-adult control group (YC) performed similarly on most of the eye movement measures, and were not correlated with classical measures of PD severity (e.g., Unified Parkinson's Disease Rating Scale (UPDRS) score).

**Discussion:** Our results suggest that PD may be associated with impairment not only in saccade inhibition, but also in binocular coordination during pursuit, and these aspects of dysfunction may be useful in PD diagnosis or tracking of disease course.

Corresponding author
Arash Yazdanbakhsh,
yazdan@bu.edu

## INTRODUCTION

Parkinson's disease (PD) is a complex neurodegenerative disorder characterized by the cardinal motor signs of tremor, rigidity, bradykinesia, disordered gait, balance, and posture (*Massano & Bhatia, 2012*; *Rodriguez-Oroz et al., 2009*). Individuals with PD have difficulty in moving the limbs and trunk, as well as in controlling oculomotor function. It has been known for some time that those with PD show prolonged saccadic latency (*Rascol et al., 1989*) and reduced smooth pursuit gain relative to healthy age-matched

individuals (*White et al., 1983*). *Uc et al. (2006)* found visual search performance was impaired in PD during driving, and *Toner et al. (2012)* showed that impaired visual search in PD could be attributed to sensory deficit. Many other studies have also reported perceptual and visuospatial disturbance in PD (*Bodis-Wollner, 2003*; *Cronin-Golomb, 2010*; *Davidsdottir, Cronin-Golomb & Lee, 2005*; *Davidsdottir et al., 2008*; *Diaz-Santos et al., 2015*). It is still unclear whether these oculomotor symptoms may co-occur with, or even arise before, the disease-characteristic motor symptoms in limbs and trunk.

Among the oculomotor functions, smooth pursuit eye movements serve an important role in vision by maintaining fixation on the selected moving object as it moves across visual field. Conventionally, smooth pursuit has been studied using small targets that produce either local or global motion signals to initiate and maintain pursuit (*Santos, Gnang & Kowler, 2012*). Although there are higher-level signals such as prediction, anticipation, and symbolic cues that may elicit anticipatory pursuit (*Kowler et al., 2014*), these high-level signals are not sufficient to maintain pursuit after its initial onset. Unlike other types of eye movements such as saccades, the quality of smooth pursuit highly relies on motion signals. The dependence of smooth pursuit on motion perception highlights the connection between the oculomotor and perceptual systems. Accordingly, the pattern of eye movements may reveal the quality of perception. Support for this possibility is provided by the existence of brain regions (such as middle temporal area (MT)) that are important for both motion perception and the generation of smooth pursuit (*Keller & Heinen, 1991*). Prior studies have shown that lesions of MT led to impairment in pursuing a moving target (*Newsome & Wurtz, 1985*), which suggests that the motion signal processed in MT is also supplied to the pursuit system. Hence, studying smooth pursuit eye movements could be a useful tool for understanding the disorders that feature both perceptual and motor dysfunction, such as PD.

Studies of oculomotor control in PD have often focused on smooth pursuit without considering the effect of saccades occurring during the pursuit (*Keller & Khan, 1986*; *Ladda et al., 2008*), or on saccade tasks in which no smooth pursuit was initiated (*Chan et al., 2005*; *Crawford, Henderson & Kennard, 1989*). When analyzing smooth pursuit, saccades are usually discarded or replaced with the results of linear interpolation (*Ke et al., 2013*). By contrast, a number of studies have pointed to the interrelation of smooth pursuit and saccades. *Erkelens (2006)* asked observers to make a saccade and then engage in pursuit of a series of moving targets in various locations. When the targets appeared one at a time, pursuit latencies were shorter than saccade latencies. But when a new moving target appeared before the currently pursued target was removed, pursuit and saccadic latencies became similar. This result raises the possibility that pursuit eye movements and saccades share a single preparatory input and may be governed by a common decision process (*Joiner & Shelhamer, 2006*; *Krauzlis & Miles, 1996*; *Krauzlis, Zivotofsky & Miles, 1999*). *Krauzlis & Dill (2002)* further found that, in superior colliculus, the activity of the same set of neurons represents target selection not only for saccades but also for smooth pursuit. Behavioral studies have also indicated a tight correlation between saccades and smooth pursuit in predicting object motion in the natural environment (*Diaz et al., 2013*). Together these findings suggest that investigating smooth

pursuit without taking into account the effect of saccades may miss the opportunity to evaluate their coordination, which may itself serve as a new measure of eye movement disturbance in PD. It may also be important for understanding of the observed changes in motor behaviors that depend upon the integration of visual signals, such as noted above for visual search while driving, and further for daily locomotion (including avoidance of falls).

Prior studies have investigated the effect of PD on saccades since one of the hallmarks of PD is the dysfunction of the basal ganglia, which also control the generation of saccades (*Bergman et al., 1998*; *Hikosaka, Takikawa & Kawagoe, 2000*). *Clark, Neargarder & Cronin-Golomb (2010)* found that observers with PD had impaired performance on an antisaccade task but not on prosaccades (reflexive saccades), as predicted by the dependence of antisaccades on frontal-lobe function, which is compromised in PD. *Helmchen et al. (2012)* studied the effect of PD on both pursuit and saccades. They found that the ability to anticipate future events before pursuit initiation was impaired in PD, but the latency of saccades did not differ from that seen in the control group. Although both saccades and smooth pursuit were investigated in this study, they were tested in separate tasks and therefore it is still unclear how well saccades are inhibited during pursuit or how the saccades are triggered.

In addition, very little research has investigated the effect of PD on binocular coordination. Binocular coordination keeps the lines of sight from two eyes aligned for the process of fusion. It has been established that individuals with PD suffer from several visual deficits, such as diplopia (*Armstrong, 2011*). It is possible that individuals with PD may also have impaired binocular coordination which leads to a deficit of convergence, as prior study found that children with dyslexia also show the deficit of binocular coordination (*Kirkby et al., 2011*). Evaluating binocular coordination in PD may provide useful insight into any dysfunction of oculomotor control.

The goal of the present study was to understand the eye movement control during smooth pursuit in PD and investigate the possible effect of PD on binocular coordination. To do this, we used a simple pursuit task while varying the viewing conditions in which the moving target could be seen by both eyes or by only one eye. This prusit task consisted of two periods (fixation period then pursuit period), which required observers to maintain their gaze first on the stationary fixation and later on a moving target. During the fixation period, the later pursuit target would keep approaching observers' gaze position so that it would serve as a distractor to increase the difficulty of holding their fixation. Normal oberservers should be able to maintain their fixation even when the distractor was heading to their gaze and then keep their fixation on the moving target during the pursuit. We assessed the observers' ability to inhibit saccades both during fixation and duing the pursuit within the same task. Aging has been found to have effects on the gain of pursuit (*Moschner & Baloh, 1994*), the dynamics and metrics of saccades, such as peak velocity and saccadic duration (*Munoz et al., 1998*), and also the inhibitory control of saccades (*Butler, Zacks & Henderson, 1999*). Considering the effects of aging motivated a novel aspect of the present study of participants with PD and age-matched normal control adults (NC) by the inclusion of a young-adult control

group (YC) in order to investigate whether aging has an effect on the control of eye movements. We were also interested in examining whether aging affects the coordination between saccades and smooth pursuit eye movements during pursuit.

## METHODS

### Participants

Thirty observers participated in the study. Ten (eight men and two women) had been diagnosed with idiopathic PD without dementia (mean age 64.5 years (SD = 7.2), mean education 17.5 years (SD = 2.4)) and another 10 (five men and five women) were healthy NC (mean age 61.2 (SD = 7.3), mean education 16.3 (SD = 1.8)). All participants with PD were screened for dementia using an extensive neuropsychological assessment that assessed multiple cognitive domains. The NC group was matched for age and education to the PD group (age, $t(18) = 1.01$, $p = 0.32$; education, $t(18) = 1.28$, $p = 0.22$). A further 10 observers were young control adults (YC, seven men and three women), all of whom were undergraduates at Boston University (age range 18–22). Members of the PD and NC groups were assessed for overall mental status using the modified Mini-Mental State Examination (MMSE; score converted to standard MMSE) (*Stern et al., 1987*). Mean MMSE for those with PD was 28.8 (SD = 0.7) and for NC was 29.2 (SD = 0.5). Participants with PD were recruited from Parkinson Disease Clinic at the Boston Medical Center and the Fox Foundation Trial Finder, and NC were recruited from the community. The PD and NC participants were compensated for their time. The YC participants received course credit. All procedures were approved by the Boston University Charles River Campus Institutional Review Board (1204E), and consent was obtained according to the Declaration of Helsinki.

Diagnosis of idiopathic PD was made by the participants' neurologists, using United Kingdom Parkinson's Disease Society Brain Bank clinical diagnostic criteria (*Hughes et al., 1992*). They met criteria for mild to moderate stages of the disorder (stages 1–3 on the Hoehn and Yahr scale (*Hoehn & Yahr, 1967*)). Disease severity was determined with the use of the Unified Parkinson's Disease Rating Scale (UPDRS, four sections; *Fahn, Elton & UPDRS Development Committee, 1987*; *Levy et al., 2005*; Table 1).

### Apparatus

Movements of the left and right eyes were tracked and recorded using an SR Research Eye-Link II head mounted eye tracker, sampling at 250 Hz. A chin rest was used to stabilize the head. The stimulus was presented by a 3D projector (Optoma HD25, HD, 1080p, 2000 ANSI Lumens, China) with a frame rate of 120 Hz through a polarized active shutter (Smart Crystal Pro, Intelligent 3D Polarization Modulator, Paris). The active shutter polarization oscillated at 120 Hz in synchrony with the projector frame update and was placed in front of the projector. Through this method, the left and right eye images could be assigned to different polarizations at each consecutive time frame. Observers wore passive polarizing glasses (Volfoni Intelligent 3D Eyewear, VP-101, Paris) throughout the experiment and the eye movements were recorded through the glasses. Observers saw only the left eye frame with the left eye for 8.3 ms and then saw only the right eye frame with the

**Table 1 Participant characteristics of individuals with Parkinson's disease.**

| Subject | Age | Gender | Education (years) | Disease duration (years) | Far acuity (4 m) | MMSE | UPDRS motor | UPDRS total | Hoehn and Yahr stage | LED (mg per day) | Duration between tests of eye movements and UPDRS (years) |
|---|---|---|---|---|---|---|---|---|---|---|---|
| PD1 | 71 | F | 18 | 9.4 | 20/25 | 29.2 | 5 | 21 | 1.5 | 770 | <0.25 |
| PD2 | 64 | M | 16 | 10.3 | 20/20 | 27.7 | 34 | 44 | 3 | 750 | <0.25 |
| PD3 | 56 | M | 12 | 3.1 | 20/16 | 28.7 | 17 | 27 | 2.5 | 225 | <0.25 |
| PD4 | 71 | M | 18 | 2.7 | 20/16 | 27.7 | 14 | 35 | 1.5 | 225 | <0.50 |
| PD5 | 73 | M | 21 | 5.2 | 20/20 | 28.7 | 9 | 20 | 1 | 10 | NA |
| PD6 | 69 | M | 19 | 2.1 | 20/16 | 29.2 | 4 | 10 | 1 | 100 | <0.25 |
| PD7 | 59 | M | 16 | 1.3 | 20/25 | 28.7 | 19 | 36 | 2 | 600 | <0.25 |
| PD8 | 64 | M | 18 | 21 | * | * | 39 | 63 | 3 | 1,675 | <0.25 |
| PD9 | 67 | M | 18 | 1.3 | 20/20 | 29.7 | 33 | 48 | 2 | 0 | <0.25 |
| PD10 | 51 | F | 19 | 1.1 | 20/25 | 29.2 | 16 | 28 | 3 | 0 | <0.25 |
| Group Mean, SD** | 64.5 (7.2) | | 17.3 (2.4) | 5.7 (6.3) | 20/20 (20/16–20/25) | | 14.8 (12.4) | 25.4 (15.5) | 2 (1.5–3) | 382.1 (530.8) | <0.5 |

**Notes:**

MMSE, mini-mental state examination; UPDRS, unified Parkinson's disease rating scale; LED, levodopa equivalent dose of medication, per Tomlinson conversion formula (*Tomlinson et al., 2010*).

* Formal acuity assessment not available. The observer was able to easily see, detect, and locate the fixation point and moving target.

** Means, SDs except for Snellen acuity (for those available) and Hoehn and Yahr scale, which are medians and ranges.

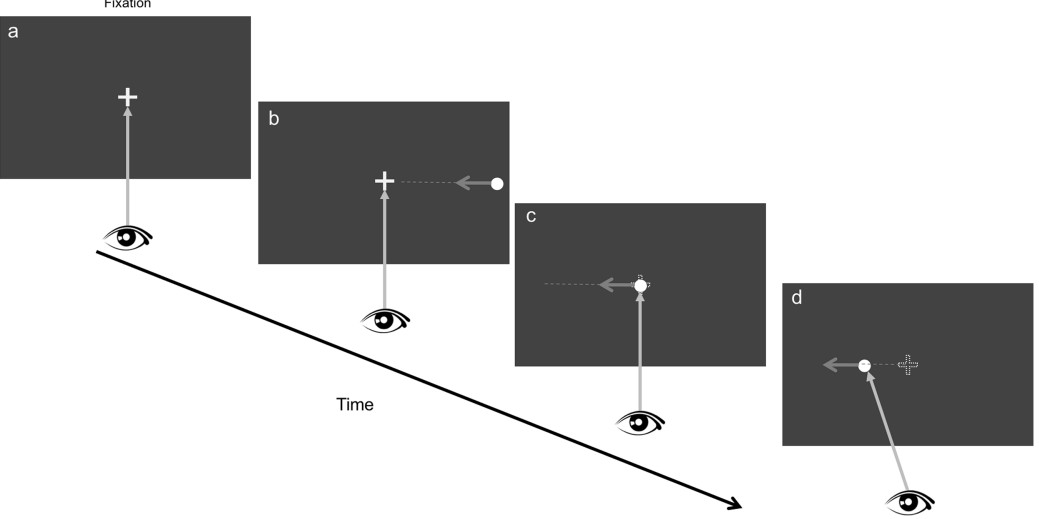

**Figure 1 Stimulus display and procedure.** (A) Fixation period (200–400 ms, randomly chosen to avoid observer's anticipation of target appearance). (B) A target appears in the peripheral visual field (e.g., at the right edge of the display) and starts to move toward the fixation point; (C) at the trigger moment, the fixation point for each eye disappears and the target keeps moving in the same direction at the original speed. (D) The observer's task is to maintain fixation during the fixation period and to pursue the moving target once it hits the central fixation point.

right eye for the next 8.3 ms. In this way, we achieved stereoscopic and dichoptic displays with passive polarizing glasses while avoiding interruption of the eye tracking signals.

## Stimulus display and procedure

The horizontal and vertical spans of the projected display on the screen were 47° and 27°, respectively. Two viewing conditions were tested separately. Each condition started with a five-point calibration procedure, followed by presentation of a small fixation cross in the center of the screen. Then observers would press the space bar to start the trial. In binocular viewing conditions, both eyes saw identical stimuli. In dichoptic viewing conditions, one eye would see the moving target and the selection of which eye seeing the target was randomly chosen. Figure 1 shows the experimental procedure in the binocular condition. Observers were instructed to fixate on a marker after the trial began (the central cross in Fig. 1A). A chin rest was used to maintain head stability. After a short duration, randomly selected in a range of 200–400 ms, a target, which was a red disc with the diameter of 0.5°, appeared at one of two possible locations (left/right edge) in the peripheral visual field (with 22° of eccentricity) and started to move toward the fixation point with the speed of 10°/s. Observers were instructed to keep fixating on the central marker (Fig. 1B) before the target hit the marker. When the target reached the central marker (trigger moment), the marker would disappear and the target kept moving along the same direction. Observers were asked to pursue the moving target as soon as the target reached the central fixation. As the target moves with a constant speed, the pursuit start time would be predictable and observers would keep pursuing the target until it reaches the edge of the display (22°). To familiarize observers with the task, we conducted
a set of practice trials before the experimental trials. After the practice trials, observers conducted 10 trials in the binocular viewing condition, followed by 20 trials in the dichoptic viewing condition. Each trial lasted approximately 6 s.

### Data analysis

Gaze positions for individual trials were stored for offline analysis. Horizontal eye velocity was calculated from the time course of horizontal gaze positions. Each velocity sample is the slope of the regression line of the gaze position samples within a sliding window of 100 ms (which is similar to *Santos, Gnang & Kowler (2012)* but the current study had a wider window). Saccade onsets and offsets were detected offline using the Eyelink velocity algorithm with a minimum amplitude criterion (1.5°). Saccades were excluded (along with blinks) in the velocity trace and pursuit velocity analysis, and the gaps were filled by interpolating the data adjacent to the gaps. Pursuit velocity was analyzed during the steady state pursuit, which was the interval 500–700 ms after pursuit starting point (*Heinen, Jin & Watamaniuk, 2011*). To avoid artifacts due to the edge of the display, the analysis focused on the 3-s interval from one and half second before to one and half second after the moving target reached the central fixation marker.

## RESULTS

Figures 2A–2C show the mean horizontal eye offset from the ideal fixating position and Figs. 2D–2F show mean horizontal eye velocity over time. In Figs. 2A–2C, the gaze position relative to the ideal fixating position is shown as a function of time, with zero indicating the moment of the target crossing the initial gaze position (trigger moment) and observers needing to start pursuing the target. Observers fixated on the central fixation point and started pursuing the horizontal moving target at time zero. Thus, the eye offset before time zero is the gaze position relative to the fixation point and the offset after time zero is the gaze position relative to the moving target. Negative numbers on position difference (vertical axis) indicate that the gaze positions were behind the target position, and positive numbers indicate that the gaze positions were ahead of the moving target. The intervals containing saccades during the fixation or smooth pursuit were excluded.

After the pursuit start time (trigger point, time = 0), there was a lag behind the target for all YC, NC, and PD. Even though the eye-fixation offset for PD was similar to NC and YC during the fixation period (negative time), the eye velocity trace became unstable for PD during fixation. By contrast, both NC and YC were able to keep their fixation on the fixation point. This pattern can be shown by the number of saccades generated during the fixation period in the later analyses.

To evaluate the quality of pursuit performance, we computed the gain of smooth pursuit, which was the average pursuit velocity during the steady-state pursuit phase (the interval 500–700 ms after pursuit starting point, see Data Analysis section), divided by the target velocity. Figure 3 shows the pursuit gain for all three subject groups.

A one-way ANOVA revealed a main effect of participant group ($F(2,27) = 6.81$, $p = 0.004$, $\eta_p^2 = 0.3$). However, a post hoc Tukey test showed no difference either between PD and NC ($p = 0.26$) or between YC and NC ($p = 0.11$).

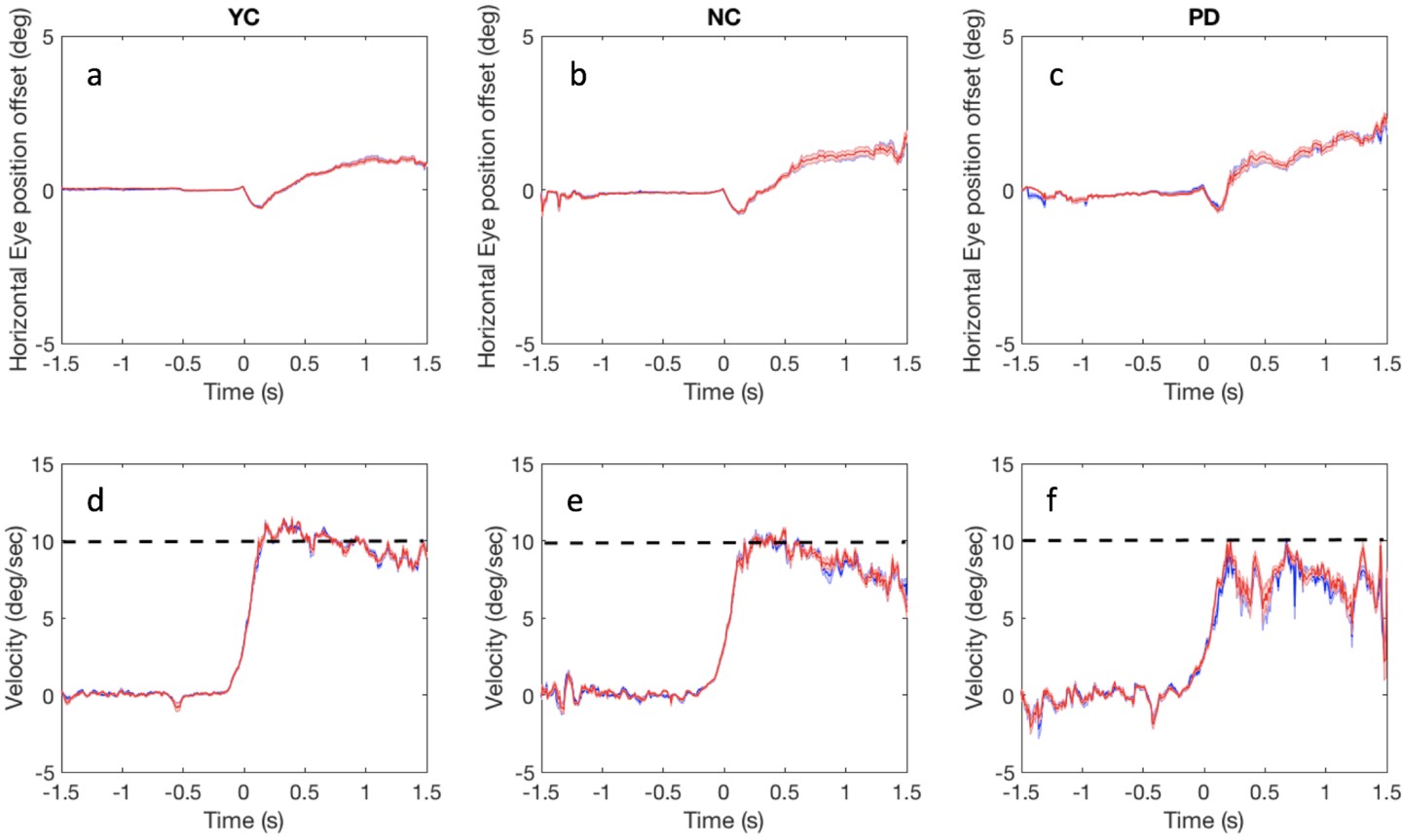

**Figure 2 Eye-fixation offset and gaze velocity.** (A–C) Show the average horizontal eye-fixation offset from the requested central fixation/the moving target and (D–F) represent the average eye velocity over time in the blank background condition for YC (A, D), NC (B, E) and PD (C, F). (A–C) Show the gaze offset from target and (D–F) show the eye velocity. The offset before time = 0 is the eye-fixation difference. The offset after time = 0 is the eye-target offset. Observers started to pursue the horizontally moving target at time = 0. The black dashed line in D–F represents target velocity (10°/s). Red and blue lines represent data from the right and left eyes, respectively. Shading indicates +/− standard error across all 10 observers per group.

Figure 4 shows the average saccade rates during fixation (−1.5 to 0 s) and during smooth pursuit (0–1.5 s) for all PD, NC, and YC participants. A one-way ANOVA indicated that during the pursuit, the saccade rates were different across subject groups ($F(2,27) = 7.2$, $p = 0.003$, $\eta_p^2 = 0.35$). A post hoc Tukey test indicated that saccade rates were different between PD and NC ($p = 0.03$) and there was no difference between NC and YC ($p = 0.69$). The saccades were analyzed separately durig pursuit initial phase (0–500 ms) and during steady phase (500–1,500 ms) and PD participants made more saccades than NC and YC in both intial phase ($F(2,27) = 4.73$, $p = 0.017$, $\eta_p^2 = 0.26$) and the steady phase ($F(2,27) = 7.08$, $p = 0.003$, $\eta_p^2 = 0.34$). During fixation, there was no significant difference found in saccade rates ($F(2,27) = 2.75$, $p = 0.08$, $\eta_p^2 = 0.17$).

The results above show that individuals with PD had higher saccade rates during the pursuit. It is possible that the increasing saccade rate during the pursuit was an attempt of correcting offset between fixation and the moving target as some prior study has shown (*Stuart et al., 2014*). To evaluate this possibility, we compared the offset error before

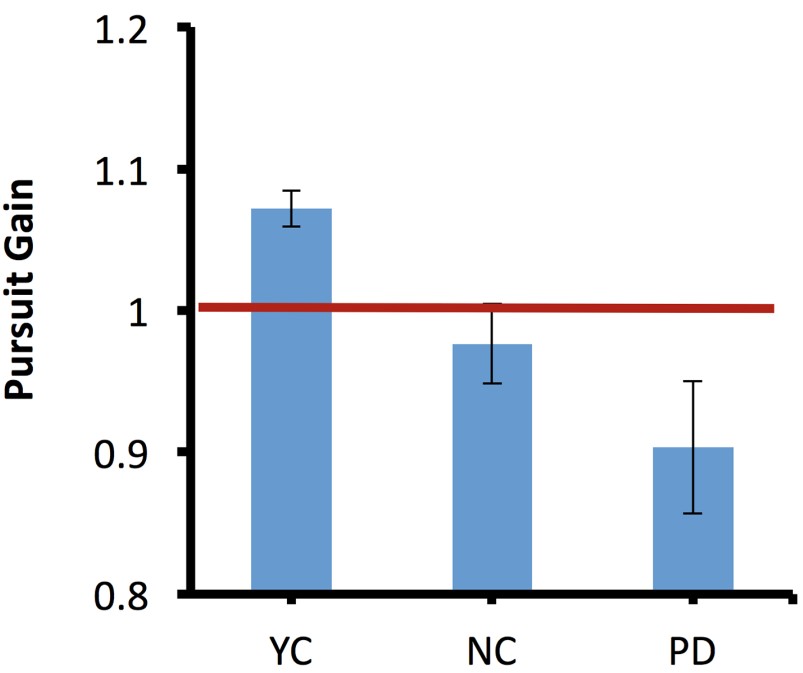

**Figure 3 Pursuit Gain in binocular viewing condition.**

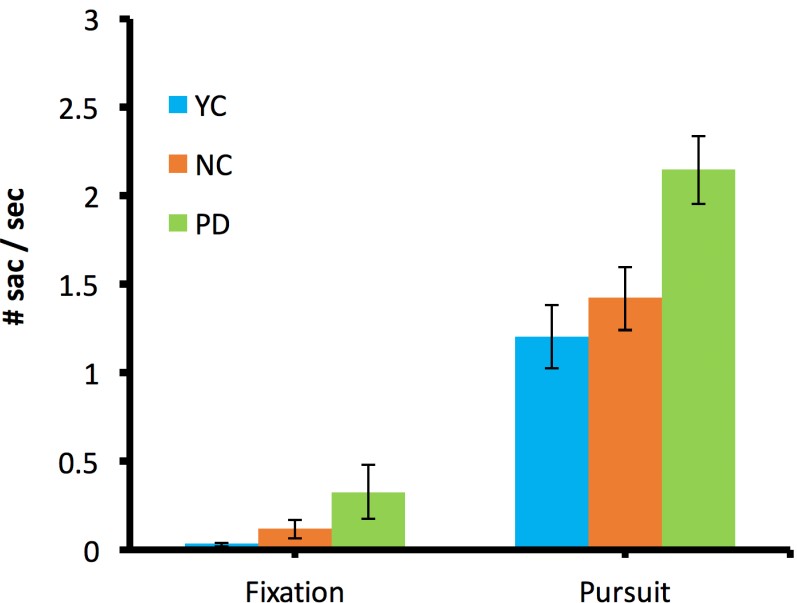

**Figure 4 Saccade rates.** Average saccade rates during fixation (−1.5 to 0 s) and smooth pursuit (0–1.5 s) for the YC (blue), NC (red), and PD (green) groups in the binocular viewing condition. The error bars indicate +/− standard error. A value of 0 in fixation indicates that there was no saccade during the fixation period.

and after saccades during the pursuit. Figure 5 shows the post- vs. pre-saccadic offset for each observer in all three groups. The offset was calculated by subtracting the target position from the eye position. Thus, the negative offset means the eye was behind the target and positive offset means the eye was ahead of the moving target.

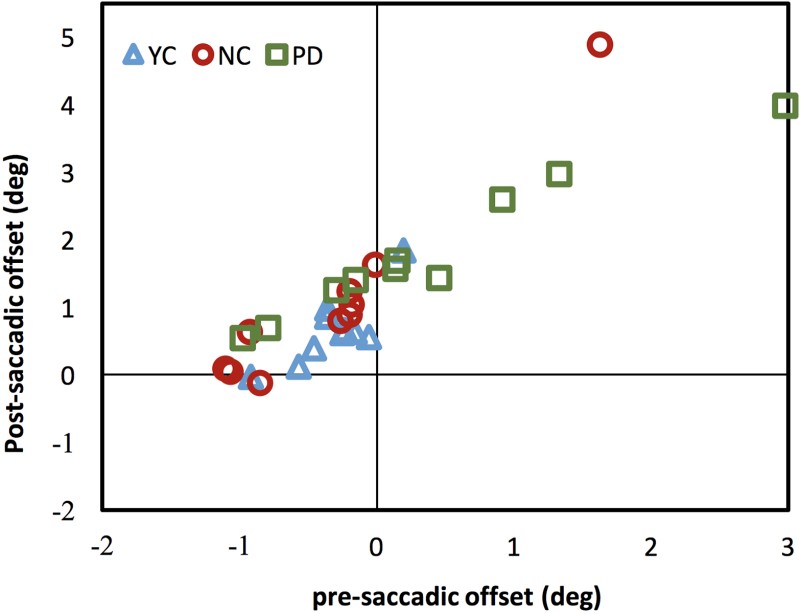

**Figure 5 Post- vs. pre-saccadic eye-target offset.** Post- vs. pre-saccadic offset for the YC (blue), NC (red) and PD (green) groups in the binocular viewing condition. Each data point represents the average data for an individual observer.

There are two possible reasons for generating saccades during pursuit: (1) observers produced catch-up saccades to compensate for the offset when the eye was behind the moving target; (2) observers failed to inhibit saccades when there was no need to initiate them. Figure 5 shows that, unlike most of NC and YC who made saccades when their fixations were behind the moving target in attempt to catch up with the moving target, about half of PD observers made the saccades when the eyes were not behind the target or even when the eyes have been ahead of the target and the saccades they made did not reduce the overall offset from the moving target. This implies that the high saccade rates in PD (Fig. 4) were not due to the attempt of correcting the pursuit error, but due to the other factors.

Taken together, these results suggest that participants with PD were more likely to have difficulty in maintaining their fixation on the moving target during pursuit. This deficit was not due to aging since the saccade rates and the cause of making saccades for the NC group were indistinguishable from those of the YC group.

## Binocular coordination

To test the possible deficit of binocular coordination, we analyzed the difference between left–right gaze positions during a 3-s interval from 1.5 s before to 1.5 s after the moving target reached to the fixation. Figure 6 shows the left–right eye divergence (the absolute difference of the left and right eye gaze positions on the screen).

While the vergence during fixation did not show a large difference for any of three groups, about half of PD participants showed an overall increasing cross-ocular difference when they started to pursue the moving target. To quantitatively examine the left–right
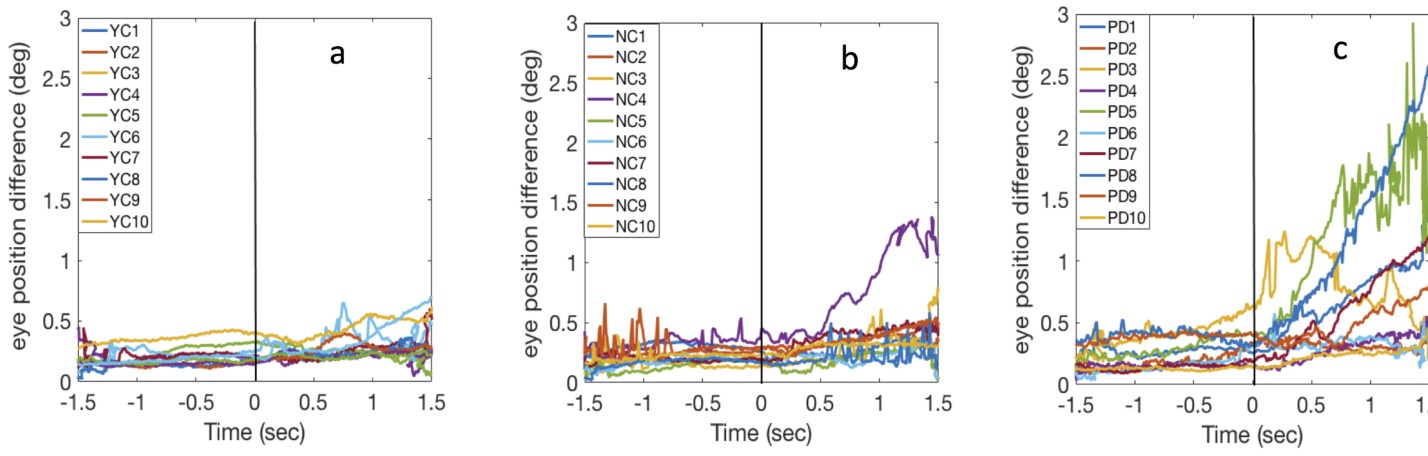

**Figure 6 Binocular divergence in binocular condition.** Left–right gaze position difference over time in the binocular blank background condition for YC (A), NC (B), and PD (C). Each line represents the averagedata from a single observer. Observers started pursuit at time = 0.

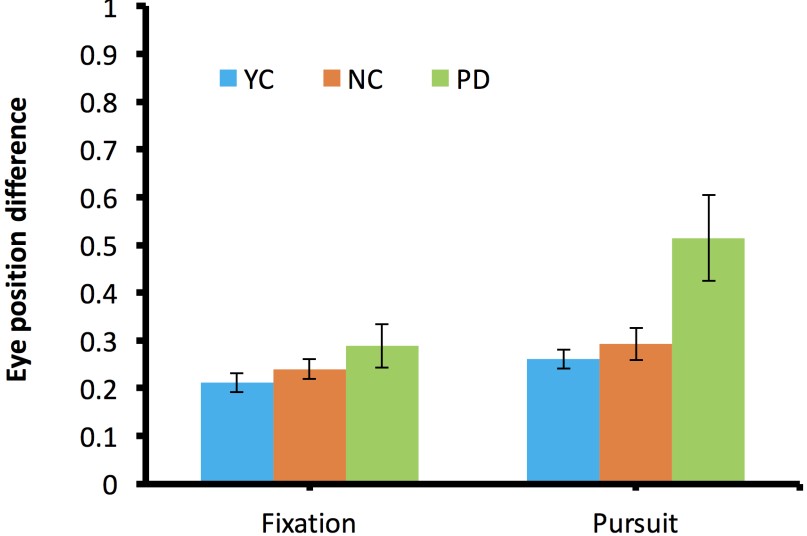

**Figure 7 Left–right eye divergence during fixation and during pursuit.**

eye divergence, we compared the average gaze position differences before and after the start of pursuit. Figure 7 shows the average binocular eye position difference during fixation and during pursuit.

During fixation, there was no difference in binocular divergence ($F(2,27) = 1.51$, $p = 0.24$). Of note, during pursuit, there was a significant difference in binocular divergence across different groups ($F(2,27) = 5.82$, $p = 0.008$, $\eta_p^2 = 0.3$). Post hoc testing revealed a significant difference between PD and NC ($p = 0.03$) but no difference was found between NC and YC ($p = 0.92$).

Within the PD group, we also examined whether the impairment of binocular coordination was related to any particular clinical characteristic, in order to establish whether our measures were sensitive to more severe disease. To do this, we computed

the Spearman correlation between the eye position difference during pursuit and scores on the UPDRS, the standard clinical measure of PD severity, including motor and other aspects of PD; the Hoehn and Yahr index of stage of disease; and the Columbia Modified MMSE, a measure of overall cognitive status (Table 1). The MMSE had a very restricted range in our sample, who were not demented. There was no correlation between eye position difference and UPDRS total score ($\rho = -0.09$, $p = 0.8$), UPDRS motor score ($\rho = 0.12$, $p = 0.73$), Hoehn and Yahr stage ($\rho = -0.16$, $p = 0.7$), or MMSE ($\rho = 0.13$, $p = 0.72$). The lack of correlation between UPDRS score with gaze position difference is comparable to what has been found in previous studies (*MacAskill et al., 2012*; *Stuart et al., 2017*). In addition, no correlation was found between eye gaze position difference and saccade rates ($\rho = 0.07$, $p = 0.85$) or between disease duration and saccade rate during pursuit ($\rho = -0.22$, $p = 0.54$) or between disease duration and other eye movement characteristics ($\rho = 0.19$, $p = 0.6$ for pursuit gain; ($\rho = -0.24$, $p = 0.51$ for eye position difference).

Although we did not find that PD pursuit gain was significantly smaller than NC, our results show that PD made more saccades than NC or YC during pursuit but not necessarily due to the need of correcting offset error during the pursuit. Moreover, binocular divergence for PD increased over time during pursuit, which was not the case for NC or YC.

## Dichoptic conditions

As shown above, it is possible that the coordination of the eyes may be impaired in individuals with PD and this could explain why we saw an increasing position difference between the eyes over time once the PD participants started the pursuit. If this is the case, we may see a similar or even larger deviation between the two eyes when each eye sees a different image. To investigate this possibility, the same observers participated in an experiment composed of trials with a dichoptic condition in which one eye could see only the moving target and the other eye could see only the background. Likewise during the fixation period (before observers start pursuing while the gaze is on the fixation point), the dichoptic condition (in which only one eye can see the fixation point), may reveal binocular divergence during the fixation period.

Figure 8 shows the difference between left and right gaze positions for the dichoptic condition. Similar to what we found in the binocular condition (Fig. 6), many PD showed an increase in the binocular divergence after pursuit was initiated, whereas most of NC and YC remained at the same level (except NC4).

Figure 9 shows the average binocular eye position difference during fixation and during pursuit. Similarly, there was no binocular divergence difference across subject types during fixation ($F_{(2,27)} = 1.63$, $p = 0.2$, $\eta_p^2 = 0.11$), but there was a significant position difference during pursuit ($F_{(2,27)} = 3.4$, $p = 0.048$, $\eta_p^2 = 0.2$). Nevertheless, Tukey post hoc testing showed only nonsignificant marginal difference in binocular divergence between PD and NC ($p = 0.07$), or between PD and YC ($p = 0.1$). A repeated measure ANOVA also shows that there was no difference in eye position difference between binocular and dichoptic viewing conditions ($F_{(1,27)} = 2.72$, $p = 0.11$).

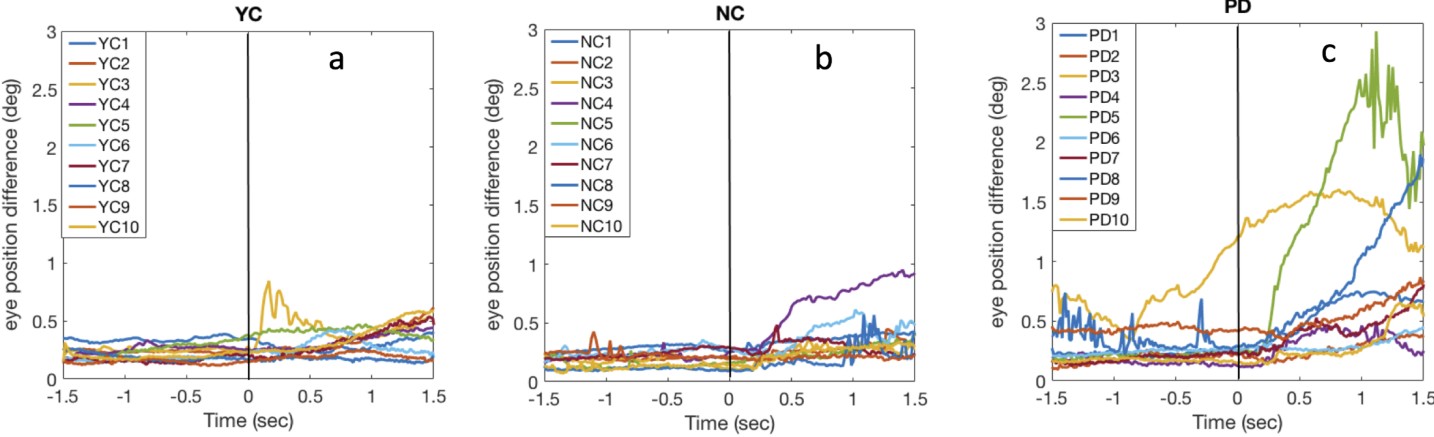

**Figure 8 Binocular divergence in dichoptic condition.** Binocular divergence over time in the dichoptic condition for YC (A), NC (B), and PD (C). Each line represents the average data from a single observer. Observers maintained their fixation then started pursuit at time = 0. Observers were same as those in Experiment 1.

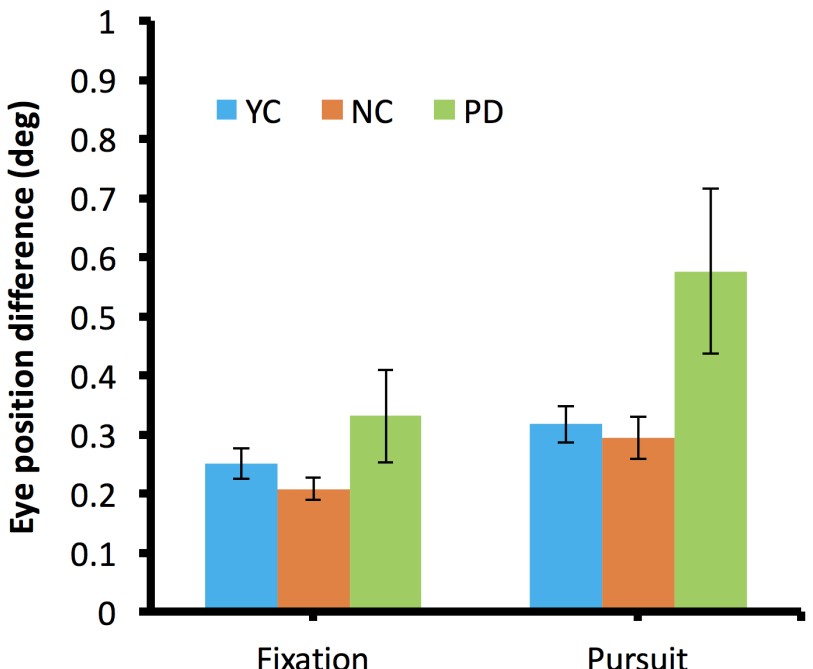

**Figure 9 Binocular divergence during fixation and during pursuit in the dichoptic viewing condition.**

The relation between binocular divergence during pursuit and the clinical characteristic were also examined. Similar to the result in the binocular condition, there was no correlation between eye position difference and UPDRS total score ($\rho = -0.006$, $p = 0.99$), UPDRS motor score ($\rho = 0.12$, $p = 0.76$), MMSE ($\rho = 0.087$, $p = 0.83$), or Hoehn and Yahr stage ($\rho = -0.18$, $p = 0.62$).

The saccade rates in dichoptic conditions were also analyzed. As found in Experiment 1, there were significant differences in saccade rates during the pursuit across groups ($F(2,27) = 4.42$, $p = 0.02$). A post hoc testing showed that there was some difference in

saccade rates between PD and NC ($p = 0.02$) and no difference was found between NC and YC ($p = 0.66$). Similarly, no correlation was found between saccade rates during pursuit and eye position difference in dichoptic condition ($r = -0.17$, $p = 0.63$).

The result of Experiment 2 shows that binocular coordination was not different across viewing conditions (binocular vs. dichoptic). That is, the eye position deviation was not larger when each eye saw different images than when both eyes saw identical images. Nevertheless, some of the participants with PD still showed an increased binocular divergence after pursuit was initiated and this increase was not seen in NC.

## DISCUSSION

The present study examined the role of saccades during pursuit and fixation in PD, in healthy adults matched to the PD group for age, and in younger control adults. Saccades are often excluded from smooth pursuit analysis as seen in many previous studies, yet they not only reveal how attention may be distributed during pursuit (*Heinen, Jin & Watamaniuk, 2011*; *Jin et al., 2013*), but also provide valuable information about the control of the oculomotor system in PD, as we show with the present experimental results. If the saccadic inhibitory system for PD is impaired (*Chan et al., 2005*; *Kitagawa, Fukushima & Tashiro, 1994*), those with PD may fail to maintain their fixation and initiate saccades while attempting to fixate a stationary point or pursue a moving target. Our results showed that individuals with PD made more saccades during pursuit (Fig. 4). These detected saccades are not simply the square wave jerks as saccade rates during fixation were too low to be conjugate. Also, the average magnitude of saccades during pursuit is about 2.4°, which is much larger than the average square wave jerks reported in previous studies (*Kapoula et al., 2013*; *Otero-Millan et al., 2013*). In addition, this frequent occurrence of saccades did not seem to be due to the effect of age since control adults who were age-matched to the PD group were able to maintain their fixation on the moving target better than PD, and their eye movements were similar to those of the young adult control group.

During pursuit, the occurrence of saccades often serves to compensate the offset error arose from the slower pursuit velocity shown in the past studies. Some observers with PD in the current study, however, made more saccades even when the fixation was not behind or even ahead of the moving target and this caused the eyes to be ahead of the moving target, as shown in Fig. 2, despite the overall lower pursuit velocity.

It is noteworthy that the increasing binocular divergence after pursuit initiation was mostly observed in PD, not in NC or YC, and this eye-position difference was similar in the binocular and dichoptic viewing conditions and was not associated with saccade rates. This suggests that the occurrence of saccades in PD obsevers may not be an attempt for correcting the fixation-target offset. Moreover, the differences were not correlated with the result of a standard motor test for PD (UPDRS). In addition, even with larger binocular eye gaze position difference, our PD observers did not report motion diplopia, which could follow the disconjugate gazes during smooth pursuit (*Kaski, Domínguez & Bronstein, 2013*). This indicates that the higher saccade rates during pursuit and the

increasing left–right eye divergence may form a new dimension independent of classical measures of PD.

The current study found that individuals with PD made more saccades during smooth pursuit than the two control groups. It is known that during pursuit, normal observers typically make catch-up saccades to decrease the gaze offset when pursuit was slower than the moving target (*De Brouwer et al., 2002*; *Collewijn & Tamminga, 1984*). In our study, unlike the two control participants who made saccades during pursuit mostly when the fixation was behind the pursuit target, more than half of the observers with PD made saccades when the fixation was not behind or even ahead of the moving target (Fig. 5). This may suggest that these saccades made by PD were not triggered by the signal of spatial offset. Instead, they may just indicate a failure of saccade inhibition.

Parkinson's disease has been associated with dopaminergic disruption in the basal ganglia, which plays a critical role in saccade inhibition (*Hikosaka et al., 1993*; *Hikosaka, Takikawa & Kawagoe, 2000*). It is possible that the depletion of dopamine in PD also affects the superior colliculus (*Basso & Evinger, 1996*). As a consequence, the inhibition signal to the superior colliculus is not sustained so that saccades are made involuntarily during fixation and during pursuit.

## Binocular coordination during pursuit

Problems in binocular coordination during reading have been found in people with dyslexia and also in people with Huntington's disease (*Bucci, Brémond-Gignac & Kapoula, 2008*; *Collewijn et al., 1988*), though little is known about this condition in PD. *Hanuška et al. (2015)* asked observers to continuously fixate a target either moving toward or moving away from them. They found that observers with PD indeed had longer latencies in their vergence eye movement and this implies a possible impairment of binocular synchronization. Our study provides a possible new insight into binocular coordination in PD. One important finding is that some observers with PD showed an increasing binocular divergence once they started to pursue, indicative of a potential deficit in binocular coordination while tracking a moving target. This result is unlikely due to measurement error because the gaze position of both eyes aligned well during fixation for all three groups. Once smooth pursuit was initiated, observers with PD were more likely to show an increased binocular positional offset over time. The same pattern of binocular offset was found across both binocular and dichoptic viewing conditions.

The original rationale for using dichoptic viewing was to further test whether binocular gaze offset would increase if the two eyes perceived different visual inputs as the larger binocular eye gaze position difference may have been resulted not only from the impaired motor control, but also from the impaired perception of visual signal. If this is the case, the manipulation of unseeing eye in the dichoptic condition would introduce larger visual disturbance and therefore may result in a bigger binocular offset. First, our result shows that in both viewing conditions, those with PD were more likely to show the binocular misalignment when they started to pursue. Second and more importantly, the gaze offset for PD in the dichoptic viewing condition is not larger than in the binocular viewing condition during the pursuit. This may suggest that the impaired coordination was not due

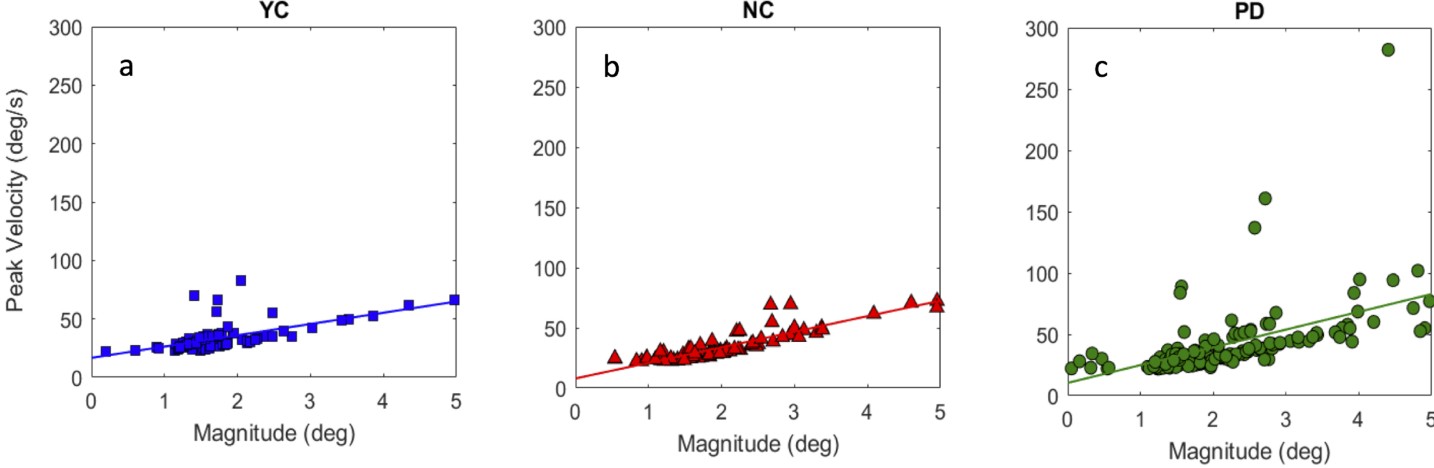

**Figure 10 Relation between saccade peak velocity and magnitude for YC (A), NC (B), and PD (C).**

to the processing of visual input but rather due to motor processing. Whether the perceived visual signal has different effects on the seeing eye and unseeing eye in PD would need further studies and investigation.

In addition, the quality of calibration may also contribute to the observed results. That is, the impaired binocular coordination may be attributed to the poor calibration in PD. Nevertheless, the quality of calibration should affect fixations, saccades and smooth pursuit in a similar way. That is, if the larger binocular eye gaze position difference was due to the poor calibration quality, we should see the similar deviation vectors with regard to disparity at any time point during the trial. To better validate the detected saccades during pursuit in the current study, we compared the relationship between saccadic peak velocity and magnitude across different groups. Figure 10 shows the pattern of saccade main sequence for each group, which indicates that saccade detection in the current study was not caused by noise. Thus, any difference in saccade rate, change in saccadic offset or binocular gaze position difference cannot simply be caused by the difference in calibration quality.

## Limits of the current study

The current study only contained limited numbers of trials and observers, and the signal noise resulted in some degree of variation as indicated in the results section. Therefore, we also presented the data of individual observers along with the main results to ensure that our readers could directly observe and access the variations of measured metrics within and across groups.

## CONCLUSIONS

The current study reexamined the ability of individuals with PD to track a moving target. We found that those with PD produced more saccades during pursuit than the control groups. We also observed that some of those with PD showed impaired binocular coordination when they started pursuit. This deficit appears to arise from abnormalities in

the oculomotor system rather than the perceptual systems, as the group differences emerged under both the binocular and the dichoptic conditions. Our measures may provide extra dimensions for PD diagnosis in addition to the standard and classic measures since the eye movement difference between the individuals with PD and age-matched NC in the present study was not correlated with classical measures of PD severity (e.g., UPDRS score or MMSE). Whether these abnormalities in eye movements occurred before the other PD-related abnormalities will need to be further investigated in the future. The current study was conducted with participants with mild-moderate PD rather than very recently diagnosed individuals because we estimated that any effect would be more likely to occur in the former group; once established, such an effect could be sought in earlier-onset cases. By pursuing these studies, we hope to achieve a better understanding of the impairment of eye movement mechanisms in PD, which may be of relevance to the understanding of oculomotor function in other neurodegenerative disorders as well.

## ACKNOWLEDGEMENTS

We would like to thank all the individuals who participated in this study. Our recruitment efforts were supported, with our gratitude, by Marie Saint-Hilaire, M.D., and Cathi Thomas, R.N., M.S.N., of the Parkinson's Disease and Movement Disorders Center at Boston Medical Center, and by the Fox Foundation Trial Finder. We also thank Sandy Neargarder, Ph.D., for helpful advice on the statistical analyses, and Sushma Hallock and Emily Fitzgerald for administrative assistance.

### Funding

This work was supported in part by grants from the National Science Foundation (NSF SBE-0354378 to Arash Yazdanbakhsh and Bo Cao) and Office of Naval Research (ONR N00014-11-1-0535 to Bo Cao, Chia-Chien Wu, and Arash Yazdanbakhsh). There was no additional external funding received for this study. The funders had no role in study design, data collection and analysis, decision to publish, or preparation of the manuscript.

### Grant Disclosures

The following grant information was disclosed by the authors:
National Science Foundation: NSF SBE-0354378.
Office of Naval Research: ONR N00014-11-1-0535.

### Competing Interests

The authors declare that they have no competing interests.

### Author Contributions

- Chia-Chien Wu performed the experiments, analyzed the data, contributed reagents/materials/analysis tools, prepared figures and/or tables, authored or reviewed drafts of the paper.

- Bo Cao conceived and designed the experiments, performed the experiments, analyzed the data, contributed reagents/materials/analysis tools, authored or reviewed drafts of the paper.
- Veena Dali performed the experiments, contributed reagents/materials/analysis tools, prepared figures and/or tables.
- Celia Gagliardi performed the experiments, contributed reagents/materials/analysis tools.
- Olivier J. Barthelemy contributed reagents/materials/analysis tools, prepared figures and/or tables.
- Robert D. Salazar contributed reagents/materials/analysis tools, prepared figures and/or tables.
- Marc Pomplun analyzed the data, contributed reagents/materials/analysis tools, authored or reviewed drafts of the paper, approved the final draft.
- Alice Cronin-Golomb analyzed the data, contributed reagents/materials/analysis tools, authored or reviewed drafts of the paper, approved the final draft.
- Arash Yazdanbakhsh conceived and designed the experiments, performed the experiments, analyzed the data, contributed reagents/materials/analysis tools, authored or reviewed drafts of the paper, approved the final draft.

## Human Ethics

The following information was supplied relating to ethical approvals (i.e., approving body and any reference numbers):

All procedures were approved by the Boston University Charles River Campus Institutional Review Board (1204E), and consent was obtained according to the Declaration of Helsinki.

## Data Availability

The raw data are provided in the Supplemental Files.

## Supplemental Information

Supplemental information for this article can be found online at http://dx.doi.org/10.7717/peerj.5442#supplemental-information.

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
