# Peer review of "Eye movement control during visual pursuit in Parkinson’s disease"

_PeerJ, doi:10.7717/peerj.5442_

## Round 0.1 · original submission · Major Revisions

Please find the comments from 3 reviewers attached. When making your revisions, please adress the reviewers' concerns in full, particularly with respect to methodological and data analysis points.

·

Basic reporting

Grammatically sound and overall well-written.
Referencing is adequate and appropriate
The article is well-structured, and whilst the Figures are of reasonable standard, it would have been helpful to include some raw traces of the fixation and pursuit data, to better determine the 'saccades' that the authors analysed. Figure 7 is missing the units on the y axis.
There is some repetition in the discussion (e.g. lines 391-399).
The article is self-contained.

Experimental design

The rationale for the methodology used could be better described. For instance, I am not clear why the paradigm combined a fixation period followed by a moving target arising from the periphery. What is the added benefit of this design compared to a 'fixation' paradigm, followed by a second 'pursuit' paradigm? Specifically, I suspect that the arrival of a pursuit target from the periphery would have added a distractor that may interfere with fixation. If such a 'distractor' was used to introduce greater potential for saccade disinhibition, then should be explicitly mentioned.

The authors should also better define the rationale for using the dichoptic and stereoscopic displays. It seems perhaps more relevant to use this for the pursuit paradigm, so the benefit of using it for the fixation section should also be explained.

I am not convinced that the current methodology was indeed assessing any convergence, so this needs to be explained.

Validity of the findings

The key findings are:
1. Reduced pursuit gain in the PD group
2. Increased number of saccades during pursuit in the PD group -I am however not clear what these 'saccades' actually are. Their frequency would suggest that they may be square wave jerks, and thus not intrusive or affecting fixation. There is no description of the plane of the saccades, their peak velocity, or amplitude.
3. Increased inter-ocular distance (divergence) during pursuit in the PD group, that was not affected by dichoptic viewing, and was not correlated to disease severity. Did patients report any diplopia during pursuit (there is a case report of 'Motion Diplopia' in Clinical Neurology & Neurosurgery by Kaski et al. the context of ophthalmoplegia that may be of relevance for the discussion).
4. That the increase in saccades during pursuit is un-related to any phase-lag. Again, it is imperative to understand what these saccades actually look like - are they square waves or intrusive (i.e. larger amplitude) saccades?

Additional comments

Overall, the results are of interest, but the methodology needs a more robust rationale. As it currently reads, it is not clear why the paradigms were chosen, and it seems that he authors have perhaps introduced quite a few different variables here, all in the same paradigm, rendering interpretation of the findings quite challenging.

Reviewer 2 ·

Basic reporting

Well-written and logically organized.

I found the use of "involuntary" saccades in the title and abstract misleading. I much prefer the more descriptive and accurate "saccades during pursuit" (or even something like "catch-up saccades") that is used throughout most of the manuscript.

The stated interest in "age" comes out of nowhere in the Introduction (3rd to last sentence). It would be nice to tie that in somewhere earlier if possible.

Line 355: missing word. "there was difference in saccade"

The data shared are not the raw data. As I understand it, the full raw data are 30 6-second trials for 3 groups (PD, NC, YC). The raw eye position data (e.g., Fig. 6) can and should be easily shared, preferably in a .mat Matlab format but even in Excel would suffice.

Fig 1 should be black and white in my opinion. Color does not seem necessary.

Fig 2: Top row need a y-axis label, not just a unit ("degrees"). e.g., Horizontal position offset? Also, would be useful to put a label for each column in the top row (YC, NC, PD).

Experimental design

Excellent motivation and research question.

Eyetracker calibration info should be included in the methods. The calibration issue is the most important and troubling aspect of this manuscript. Potential differences in eyetracker calibration between subject groups are not addressed until the very end of the manuscript. Calibration would not affect fixation and pursuit equally as the authors claim because the physiological demands of each are different. Pursuit is driven by retinal slip so 1-2 deg of visual angle of slip away from the pursuit target would elicit pursuit while fixation would largely be unaffected by 1-2 deg of movement within a fixation window, which is a typical fixation window size. (The authors do not mention any restrictions or viewing windows for their task).

This phrasing initially made me think that the 10/20 trials were *practice* trials, which they are not. Consider improving the language for clarity: "To familiarize observers with the task, we conducted a set of practice trials before the experimental trials. There were 10 trials in the binocular viewing condition and 20 trials in the dichoptic viewing condition. Each trial lasted approximately 6 seconds."

What exactly is steady state pursuit? I see that a Heinen paper is referenced for the interval used (500-700 ms) but groups like Osborne and Lisberger labs tend to use some "initiation/open loop" interval from ~0-150 ms and steady state presumbly begins after that initiation period. Why the large discrepancy? Does this matter?

How exactly were saccades excluded? Those trials were discarded or the data were interpolated?

Validity of the findings

Line 233: Sentence started "Even though..." could use a statistic or some quantification to support this claim.

Line ~266: Could you make a polar plot of the direction of PD saccades to demonstrate your point?

·

Basic reporting

no comment

Experimental design

no comment

Validity of the findings

See comments below.

Additional comments

The manuscripts studies eye movements in PD patients during pursuit. It shows that there are more saccades during pursuit in PD than in controls and that there is more disparity between the left eye and the right eye in PD than in controls. These results are interesting and worth publishing. However, with the current analyses is not possible to discard the possibility that some of the results are not due to poorer quality recordings in the PD patients than in controls. A few additional analysis could help solidify this point.
Major comments:
- The discussion about calibration is not correct. First calibration and noise are two different things. Calibration is related to accuracy, while noise relates to precision. You could have a perfect calibration with high noise or a terrible calibration with no noise. Second, the calibration, in the simplest case will have an offset component and a gain component. It is possible to calibration a proper offset from a wrong gain. Thus, there are many scenarios that could indeed produce some of the results found in the study. It is possible and even likely that calibration and/or noise are different or each eye. For, it could be possible that both eyes have a correct offset, so fixation looks ok, but gain is wrong on one eye only so as soon as the eye moves away from center disparity increases. Eyelink eye trackers use polynomial fits for calibration that are a bit more complicated to interpret but would be susceptible to same type of problems.
- It would be helpful to show a main sequence of the saccades detected in the three populations both during fixation and during pursuit. A clean main sequence (peak velocity vs. amplitude) would be a strong indication that the saccades being detected are real and not just caused by noise.
- It is unclear how many of the saccades analyzed are saccades that occur during pursuit initiation and how many are proper involuntary saccades during pursuit. Authors may want to analyze separately the saccades that occur during the first 0.5 s of pursuit and the ones occurring after, during steady state.
- The dichoptic condition misses a key analysis. It should include a comparison of the eye that sees the target vs the eye that does not see the target. Instead of plotting eye position difference authors could plot eye to target difference. This would allow to distinguish between three possibilities. A) the eye that sees the target is pursuing well but the other eye is not. B) both eyes are pursuit poorly and independently. C) Data from one is worse not matter if it is the one seeing the target or not.

Minor:
• Did the polarizing glasses affect the eye movement quality in any way? Did the cameras of the eye tracker saw the eye through the glasses?
• Until what eccentricity did the target move during the pursuit? Also 22 degress?
• Was there any point in randomizing the latency of the pursuit start? It will still be completely predictable when they have to start pursuit after seeing the target moving toward the fixation spot.
• Figure 2: it would be clearer to show also a row with the actual average eye position together with the target position.
• Figure 3: It would be helpful to remind the reader that the gains were calculated on the 500-700 ms window after pursuit onset. Otherwise it appears to contradict Figure 2.
• Figure 4: what windows of time where used for this analysis? -1.5 to 0 and 0 to 1.5?

---

## Round 0.2 · accepted · Accept

Congratulations on your MS acceptance to PeerJ.

# ·

Basic reporting

no comment

Experimental design

no comment

Validity of the findings

no comment

Additional comments

The authors have addressed the reviewer remarks satisfactorily and I have no furtherr comments to add.

Reviewer 2 ·

Basic reporting

no comment

Experimental design

no comment

Validity of the findings

no comment

Additional comments

Much improved. All issues addressed. Good work.